# ITERATIVE SEARCH ATTRIBUTION FOR DEEP NEURAL NETWORKS

## ABSTRACT

Deep neural networks (DNNs) have achieved state-of-the-art performance in a number of application areas. However, to ensure the reliability of a DNN model and achieve a desired level of trustworthiness, it is critical to enhance the interpretability in terms of the model inputs and outputs. Attribution methods are an effective means of Explainable Artificial Intelligence (XAI) research. However, the interpretability of existing attribution algorithms varys depending on the choice of reference point, the quality of constructed adversarial samples, or the applicability of gradient constraints in specific tasks. To effectively and thoroughly explore the attribution integration paths, in this paper, inspired by the iterative generation of high-quality samples in the diffusion model, we propose an Iterative Search Attribution (ISA) method to achieve more accurate attribution by distinguishing the importance of samples during gradient ascent and descent and clipping the relatively unimportant features in the model. Specifically, we introduce a scale parameter during the iterative process to ensure that the parameters in the next iteration are always more significant than the parameters in the current iteration. Comprehensive experimental results show that our method has superior results in image recognition interpretability tasks compared with other sota baselines. Our code is available at: https://anonymous.4open.science/r/ISA-6F6B

## 1 INTRODUCTION

Deep learning networks (DNNs) have now achieved state-of-the-art performance in the tasks of computer vision (Esteva et al., 2021; Jabbar et al., 2018; Pathak et al., 2018), natural language processing (Collobert & Weston, 2008; Ozcan et al., 2021; Rajendran & Topaloglu, 2020), and speech recognition (Pan et al., 2012; Maas et al., 2017) and so on. Nevertheless, DNN models are considered as black-box approach impeding human comprehension, which may cause catastrophic decision-making. From the pespectives of model security and stability, it becomes essential to provide the interpretability of such DNN models. However, explaining the intermediate processes of model inputs to outputs is challenging due to the complexity of nonlinear layers as well as huge amount of parameters in DNN models (Pan et al., 2021).

Gradient-based attribution methods are widely used for Explainable Artificial Intelligence (XAI) as they offer an effective way to explain deep learning models. Integrated Gradient (IG) (Sundararajan et al., 2017) method proposes the axiomatic theorem of attribution for the first time and uses the reference input as an anchor on the integration path to calculate the partial derivative of the model output with respect to the input. Boundary-based Integrated Gradient (BIG) method (Wang et al., 2022) use adversarial attacks to find more close decision boundary, which helps find more faithful attribution results. Adversarial Gradient Integration (AGI) method (Pan et al., 2021) is proposed as a method of searching for adversarial samples to avoid the impact of invalid reference selection on attribution accuracy, rather than the selection of reference points. Committed to reducing the instability of IG on the integration path and mitigating the effects of noise generated in regions unrelated to the prediction category, Guided Integrated Gradients (GIG) method (Kapishnikov et al., 2021) adjusts the range of gradients by guided backpropagation to improve the accuracy of attribution.

To summarize, the literature indicates that prior researches to gradient-based attribution method often rely on baseline points selected, the quality of constructed adversarial samples, or specific gradient rules. The interpretability of existing attribution algorithms varies due to limited exploration

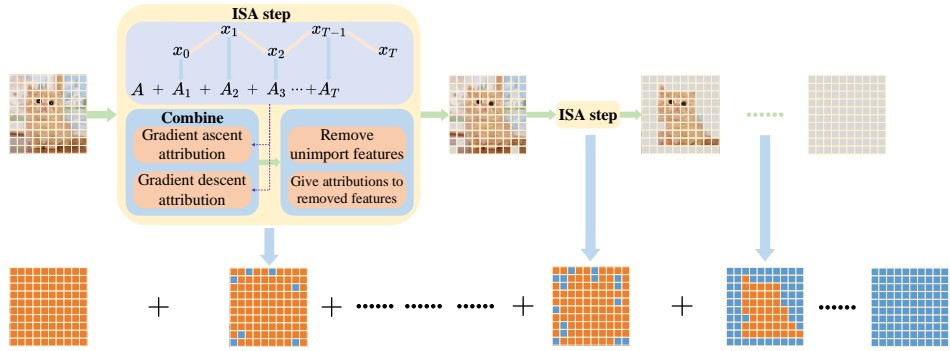

Figure 1: ISA Attribution Processing

of attribution integration paths, leading to less promising performance. In this paper, we leverage a unique search space to find the most effective features of the deep learning model and solve the problem of parallel processing of model features. Inspired by the concept of the diffusion model (Rogers, 2004), we believe that the process of iteratively computing feature importance would be more applicable for the derivation of attribution results. We firstly use both gradient ascent and gradient descent to investigate the impact of the original feature changes, which lead to the changes in outputs. Subsequently, we clip the redundant features and assign them lower attributions. Moreover, we introduce a scale parameter during the iterative process to ensure that the parameters in the next iteration are always more significant than the parameters in the current iteration, enhancing the accuracy of feature importance estimation. Figure 1 provides an illustration of our attribution processing step. More details will be provided in Section. 4.

The key contributions of this paper are summarized as follows:

- We propose a novel iterative attribution method based on gradient ascent and descent search strategies, termed ISA method, to enhance the attribution performance.

- We provide the theoretical proof and in-depth analysis for the iterative search attribution method, and perform extensive comparative and ablation experiments to evaluate the effectiveness of our method.

- We demonstrate that the ISA method can be easily implemented to obtain the state-of-art performance in comparison with other attribution methods.

## 2 RELATED WORK

Current popular methods for interpreting DNNs can be categorised into local approximation and gradient-based attribution methods (Li et al., 2023). For local approximation methods, mostly it can only interpret a single sample or a small portion of samples locally, while for gradient-based attribution methods, it tends to use gradient information to obtain a global interpretation of each feature. The latter can avoid approximation errors leading to higher attribution accuracy. Hence, in this section, we will briefly review several representative algorithms of local approximation methods and provide a more comprehensive literature review of gradient-based attribution methods.

### 2.1 LOCAL APPROXIMATION METHODS TO INTERPRET DNNS

Local approximation methods are dedicated to finding an approximately interpretable surrogate model given diverse model images to compute gradient information and obtain attribution results. The Local Interpretable Model-agnostic Explanations(LIME) algorithm (Ribeiro et al., 2016) is proposed to combine approximation and weighted sampling methods to construct a local model that gives the interpretable prediction results of the model classifier. Since the LIME algorithm is based on a locally interpretable model, it is less interpretable in the global context and has the potential to produce misleading results. The Shapley Additive Explanations (SHAP) algorithm (Lundberg & Lee, 2017) proposed by Lundberg et al. calculates the contribution of each feature to the prediction

result by Shapley value and ranks the importance, thus achieving both local and global interpretation of the model. The SHAP algorithm is able to give an adequate explanation of the model prediction in the global context compared to the LIME algorithm. However, it has a high computational cost due to the need to repeatedly calculate Shapley values for different features. Deep Learning Important Features (DeepLIFT) method (Shrikumar et al., 2017) calculates the importance score of each input feature to explain the prediction effect of the deep learning model. Some other methods including the works in (Datta et al., 2016) and (Fong & Vedaldi, 2017) are also used to obtain the interpretable results for DNNs models. However, despite the effectiveness of local approximation methods in explaining the model's local decision, they lack a global perspective on model behavior and struggle to provide promising explanations for complex and nonlinear DNN models.

## 2.2 GRADIENT-BASED ATTRIBUTION METHODS TO INTERPRET DNNS

In order to obtain reliable evaluations to cope with realistic and highly sensitive deep learning tasks, Grad-CAM (Selvaraju et al., 2017) and Score-CAM (Wang et al., 2020) use gradient information to visualize the contribution values of image pixels to explain the model prediction process. Unfortunately, these two methods are more suitable for CNNs and perform poorly in non-CNN cases. Saliency Map (SM) method (Simonyan et al., 2013) can obtain interpretable results of the model in a non-CNN environment, but suffers from gradient saturation and the attribution result may be zero.

The Integrated Gradients (IG) method (Sundararajan et al., 2017) addresses the gradient deficiency of the SM algorithm and proposes two axiomatic criteria: *Sensitivity* and *Implementation Invariance*. As mentioned in (Sundararajan et al., 2017), an attribution method satisfies *Sensitivity* if for every input and baseline that differ in one feature but have different predictions then the differing feature should be given a non-zero attribution. For two neural networks with the same inputs and outputs, the attribution is always the same if they satisfy *Implementation Invariance*. By selecting the desired reference points as anchors on a linear integration path, IG integrates the continuous gradients to obtain the attribution of each input feature. The formula of IG is expressed in Eq1.

$$IG_j(x) = (x_j - x_j') \times \int_{\alpha=0}^{1} \frac{\partial F(x' + \alpha \times (x - x'))}{\partial x_j} \mathrm{d}\alpha \tag{1}$$

where $j$ denotes the $j$-th input feature, $\frac{\partial F(x' + \alpha \times (x - x'))}{\partial x_j}$ is the gradient of model $F$ w.r.t input feature $x_j$. $x_j'$ represents the reference input feature.

Based on the exploration of better baseline selection compared to IG, the Boundary-based Integrated Gradient (BIG) method (Wang et al., 2022) introduces boundary search to obtain more accurate attribution results. Given deep learning network $F$, Integrated Gradient $g_{IG}$ and an input feature $x$, the formula is shown in Eq 2.

$$B_{\mathrm{IG}}(\mathbf{x}) := g_{\mathrm{IG}}(\mathbf{x}; \mathbf{x}') \tag{2}$$

where $x'$ is the nearest adversarial example to $x$, i.e. $c = F(x) \neq F(x')$ and $\forall \mathbf{x}_m \cdot \|\mathbf{x}_m - \mathbf{x}\| < \|\mathbf{x}' - \mathbf{x}\| \to F(\mathbf{x}) = F(\mathbf{x_m})$. Although BIG attempts to use an adversarial sample as baseline, its integration path is still linear. Meanwhile, BIG needs to calculate the gradient of each feature, which increases the computational complexity to some extent.

The Adversarial Gradient Integration(AGI) method(Pan et al., 2021) is committed to finding a steepest non-linear ascending path from the adversarial example $x_i'$ to x, which does not need reference points on the path like IG. The formula is described in Eq.3.

$$AGI_j(x) = AGI_{j-1}(x) - \bigtriangledown_{x_j} f^i(x) \cdot \epsilon \cdot sign(\frac{\bigtriangledown_{x_j} f^i(x)}{|\bigtriangledown_x f^i(x)|}) \tag{3}$$

where $\bigtriangledown_{x_j} f^i(x)$ means the gradient corresponding to false class label $i$. Step size is represented by $\epsilon$. The formula integrates along the path until $argmax_l f^l(x) = i$. It is clear that the accuracy of attribution depends on the quality of adversarial samples, and the effectiveness changes when the construct method of adversarial samples is varying.

Considering the path noise problem in the IG algorithm, the Guided Integrated Gradients (GIG) method (Kapishnikov et al., 2021) eliminates unnecessary noisy pixel attributions by constraining the network input and back-propagating the gradients of the neurons so that only the pixel attributes associated with the predicted category are retained. However, GIG is limited to the image domain

and the quality of the input features will largely affects the attribution accuracy, while the computational complexity of the algorithm is also an issue. Other variations of IG algorithm such as Fast-IG (Hesse et al., 2021) and Expected Gradient(EG) (Erion et al., 2021) have similar problems.

## 3 PRELIMINARIES

### 3.1 THE IMPORTANCE OF GRADIENT ASCENT AND DESCENT IN ATTRIBITION

The essence of gradient-based attribution algorithms, such as IG, is gradient accumulation, specifically the accumulation of gradient transformation w.r.t. each feature point between the sample $x$ to the baseline sample $x'$. BIG uses the adversarial sample as its baseline while selecting a linear path between $x$ and $x'$. AGI uses the steepest ascending process between $x$ and the adversarial sample $x'$ as the basis for gradient accumulation. Existing attribution algorithms more or less apply gradient ascent as a search means, to which we introduce gradient descent as a complementary measure, i.e., finding what features the model gets will enhance the confidence level. If making slight parameter changes results in an increase in the model's confidence, then these parameters are also considered important for the model.

### 3.2 PROBLEM DEFINITION

Formally, to explain the explicit expression of the DNN model $f(\cdot)$, we define the input feature $x \in R^n$ where $n$ is the dimension of the input feature, and the output of the model $Y = f(x)$. The goal of attribution is to find $A \in R^n$ to interpret the importance of each feature in $x$. For easy understanding, we refer to the basic idea of the Saliency Map (Simonyan et al., 2013). If the deep neural network $f$ is continuously differentiable, the input feature importance $A$ of the model will be derived from the gradient information $\frac{\partial f}{\partial x}$. It is important to highlight that this process involves a direct one-to-one mapping.

## 4 ITERATIVE SEARCH ATTRIBUTION

### 4.1 FEATURE SEARCHING IN DEEP NEURAL NETWORKS

Traditional gradient-based attribution algorithms, such as BIG and AGI, which uses adversarial samples, only consider the case where the gradient increases. In the meantime, when the goal is to explore the deep neural network space, leveraging gradient descent is also a valid approach.

We assume that the model input is $x$, then for the iterative step $t = 0, 1, ..., T$, the model gradient ascent and descent process can be expressed as

$$x_t = x_t \pm \eta \cdot sign(\frac{\partial L(x_t; W)}{\partial x_t}) \tag{4}$$

Here $\{\eta, T\} = \{\eta_1, \eta_2, T_1, T_2\}$, which value depends on whether gradient ascent or descent is performed. $\{\eta_1, T_1\}$ is step size and iterative step in gradient ascent, $\{\eta_2, T_2\}$ is step size and iterative step in gradient descent. $W$ is the parameter matrix of the model, $\frac{\partial L(x_t; W)}{\partial x_t}$ is the derivative of the loss function $L$ w.r.t. the input $x_t$. It is worth noting that in gradient ascent, $\pm$ is a plus sign. Conversely, in gradient descent, $\pm$ is a negative sign.

### 4.2 ATTRIBUTION COMBINING GRADIENT ASCENT AND DESCENT

In this work, inspired by AGI (Pan et al., 2021), we design the searching mechanism to identify the integration path instead of the original linear path. We denote $x_0$ as the original input, the path can be represented as $x_t = x_0 + \sum_{k=0}^{t-1} \triangle x_t$. In order to consider both the role of gradient ascent and gradient descent in feature attribution, we list the attribution steps below.

$$A = A + \triangle x \cdot \frac{\partial L(x_t; W)}{\partial x_t} \tag{5}$$

$$\triangle x = \pm \eta \cdot sign(\frac{\partial L(x_t; W)}{\partial x_t}) \tag{6}$$

**Presentation** Since gradient ascent implies a performance loss, we consider larger attribution results in gradient ascent to be more important because they are more likely to impact model performance. Similarly, gradient descent implies a performance enhancement. We believe that larger attribution values in gradient descent are more helpful for model performance improvement. Therefore, we should pay more attention to the small attribution results during gradient descent to evaluate whether it makes a beneficial contribution to the gradient descent process. Therefore, we can infer that in the process of *gradient ascent*, features with larger attribution values are *more important*. Conversely, features with larger attribution values are *less important* in *gradient descent*.

Since the difficulty of feature search corresponding to gradient ascent and gradient descent is different, the value of the loss function changed at the same number of iterations has a variability. We define $\triangle L_a = L(x_{T_1}) - L(x_0)$ and $\triangle L_d = L(x_{T_2}) - L(x_0)$. So we divide the attribution results in Eq.5 by the total change in the loss function to make the gradient ascent and descent equally competitive. Thus, we get $A_{ascent} = \frac{A}{\triangle L_a}$ and $A_{descent} = \frac{A}{\triangle L_d}$. Finally, we make a balance between gradient ascent and descent by combining them in the following equation

$$Attr = A_{ascent} + A_{descent} \tag{7}$$

**Discussion** It is obvious that the value of $L_a$ is positive and the value of $L_d$ is negative. As stated in the **Presentation**, the *smaller* attribution value in $A_{ascent}$ is *less important*. Since the sign of $L_d$ is negative, the *smaller* attribution value in $A_{descent}$ that was more important is transformed to be *less important*. Therefore, the *smaller* attribution value of Eq.7 is *less important* in our opinion.

### 4.3 PREREQUISITES OF ISA

**Iterative Integrity** Since the attribution values of all features need to be obtained, the iterative process has to iterate over all parameters. Suppose we have $n$ features and each time we attribute $k$ features in $x$, the total number of iterations is $\Gamma = \lceil \frac{n}{k} \rceil, \lceil \cdot \rceil$ denoted as the rounding up function.

**Feature removal priority** Features that are removed first have relatively lower attribution values than those that are removed later. In order to make the imputation results quantifiable, we perform normalization to ensure that they are between 0 and 1.

$$attr_\gamma = min_k(Attr_\gamma)$$
$$attr_\gamma = \frac{attr_\gamma - min(attr_\gamma)}{max(attr_\gamma) - min(attr_\gamma)} \tag{8}$$

We get a minimum $k$ attribution value in $Attr$ in $\gamma$ iteration of attribution as $attr_\gamma$. Such values are the removal attribution value. Following the principle that the early removal features are less important, we need to make sure $max(attr_\gamma) < min(attr_{\gamma+1})$. So that we derive

$$attr_\gamma = attr_\gamma + \gamma \tag{9}$$

For example, when $\gamma = 0$, $attr_0 \in (0, 1)$. When $\gamma = 1$, $attr_1 \in (1, 2)$. This satisfies the conditions presented above.

### 4.4 SCALING FACTOR OF ISA

We observe that the best results in the previous iteration will perform better than the worst results in the latter iteration. Thus, we add a scaling factor in the iterations to enhance the performance of the algorithm. The specific iterative formulas of ISA are as follows

$$attr_\gamma = attr_\gamma \cdot S + \gamma \tag{10}$$

where $S \in [1, 2)$ denotes the scale level of $attr_\gamma$. So we use an example to explain Eq.10. It is obvious that when $\gamma = 0$, $attr_0 \in (0, S)$. When $\gamma = 1$, $attr_1 \in (1, S + 1)$. By analogy, our assumptions are satisfied.

### 4.5 AXIOMATIC PROOF OF ISA

**Proof of Sensitivity** During the iteration process, changes in gradient ascent and descent are all captured by original input information. It is also not retroactive because the feature values in previous iterations are invariant in subsequent iterations. Thus, the attribution result must be non-zero.

**Proof of Implementation Invariance**  Since our algorithm follows the chain rule of gradients, it satisfies the requirement of Implementation Invariance in (Sundararajan et al., 2017).

---

**Algorithm 1** Iterative Search Attribution

---

**Input:** Original input feature $x_0$, parameter matrix $W$, step size $\eta_1$, step size $\eta_2$, ascent step $T_1$, descent step $T_2$, feature removal number $k$, integration step $\Gamma$, loss funtion $L$, Scaling factor $S$
**Output:** $Attr$

1: **Initial:** $Attr = 0$
2: **for** $\gamma$ in range $\Gamma$ **do**
3:     **for** $t = 0, 1, ..., T_1$ **do**
4:       $x_t = x_t + \eta_1 \cdot sign(\frac{\partial L(x_t; W)}{\partial x_t})$
5:       $A_a = A_a + \eta_1 \cdot sign(\frac{\partial L(x_t; W)}{\partial x_t}) \cdot \frac{\partial L(x_t; W)}{\partial x_t}$
6:     **end for**
7:     **for** $t = 0, 1..., T_2$ **do**
8:       $x_t = x_t - \eta_2 \cdot sign(\frac{\partial L(x_t; W)}{\partial x_t})$
9:       $A_d = A_d - \eta_2 \cdot sign(\frac{\partial L(x_t; W)}{\partial x_t}) \cdot \frac{\partial L(x_t; W)}{\partial x_t}$
10:    **end for**
11:    $\Delta L_a = L\left(x_{T_1}\right) - L\left(x_0\right), \Delta L_d = L\left(x_{T_2}\right) - L\left(x_0\right)$
12:    $A_{ascent} = \frac{A_a}{L_a}, A_{descent} = \frac{A_d}{L_d}$
13:    $Attr_\gamma = A_{ascent} + A_{descent}$
14:    $attr_\gamma = min_k(Attr_\gamma)$
15:    $attr_\gamma = \frac{attr_\gamma - min(attr_\gamma)}{max(attr_\gamma) - min(attr_\gamma)}$
16:    $attr_\gamma = attr_\gamma \cdot S + \gamma$
17:    $Attr = Attr + attr_\gamma$
18: **end for**
19: **return** $Attr$

---

## 5 EXPERIMENTS

In our study, we orchestrate an array of experiments encompassing three models, namely Inception_v3 (Szegedy et al., 2016), ResNet_50 (He et al., 2016), and VGG16 (Simonyan & Zisserman, 2014). The focal objective of these designed experiments is to discern the relative efficacy of seven distinct attribution methods, namely IG (Sundararajan et al., 2017), FastIG (FIG) (Hesse et al., 2021), GuidedIG (GIG) (Kapishnikov et al., 2021), BIG (Wang et al., 2022), SaliencyMap (SM) (Simonyan et al., 2013), AGI (Pan et al., 2021), and ISA (our work). To statistically analyse and evaluate the performance characteristics, we apply the Insertion and Deletion score (Pan et al., 2021). We demonstrate that ISA has better performance compared to other attribution methods.

### 5.1 DATASET

In the experiment, we employ the widely used ImageNet (Deng et al., 2009) dataset. We randomly select 1000 samples from ImageNet dataset to evaluate the performance of various attribution methods. The sample size is determined based on the guidelines followed by FIA (Wang et al., 2021), NAA (Zhang et al., 2022), and AGI (Pan et al., 2021) experiments.

### 5.2 EVALUATION METRICS

We follow the evaluation metrics used in AGI (Pan et al., 2021), namely the Insertion and Deletion score. The Insertion score measures the extent of output change in the model when pixels are inserted into the input. Conversely, the Deletion score quantifies the impact on the model's output when pixels are removed from the input. However, due to the adversarial nature of neural networks, the Deletion score may not always provide reliable indications (Petsiuk et al., 2018). Thus, the Insertion score offers broader and more informative insights compared to the Deletion score. This is because the Insertion score starts from a baseline of inserting crucial features. If these features are indeed

significant, the Insertion score will exhibit a rapid increase. On the other hand, the Deletion score starts from the original image, and the model can deduce partial information from the surrounding background, which cannot be controlled for the removed features. Therefore, **the Insertion score serves as a more representative indicator of the performance of attribution algorithms**.

## 5.3 EXPERIMENTS SETTING

We perform the experiments on a platform with a single Nvidia RTX3090 GPU. Meanwhile, we configure the experiment with several critical parameters. Specifically, we set the step size to be 5000, ascent step $T_1$ and descent step $T_2$ to be 8 of each, learning rate to 0.002, and $S$ to 1.1.

## 5.4 RESULT

Table 1: Insertion and Deletion sore

| Model | Method | Deletion score(mean) | Deletion score(AUC) | **Insertion score**(mean) | **Insertion score**(AUC) |
|-------|--------|---------------------|---------------------|---------------------------|--------------------------|
| Inception_v3 | IG | 0.0445 | 0.0426 | 0.3215 | 0.3208 |
| | FIG | 0.0475 | 0.0456 | 0.2029 | 0.2017 |
| | GIG | 0.0363 | 0.0343 | 0.3194 | 0.3187 |
| | BIG | 0.0557 | 0.0538 | 0.4840 | 0.4840 |
| | SM | 0.0649 | 0.0631 | 0.5331 | 0.5334 |
| | AGI | 0.0676 | 0.0658 | 0.6249 | 0.6256 |
| | ISA | 0.0745 | 0.0727 | **0.7293** | **0.7304** |
| ResNet_50 | IG | 0.0302 | 0.0283 | 0.1467 | 0.1454 |
| | FIG | 0.0342 | 0.0324 | 0.1078 | 0.1063 |
| | GIG | 0.0210 | 0.0191 | 0.1463 | 0.1450 |
| | BIG | 0.0485 | 0.0467 | 0.2911 | 0.2905 |
| | SM | 0.0585 | 0.0567 | 0.3160 | 0.3154 |
| | AGI | 0.0532 | 0.0515 | 0.5133 | 0.5136 |
| | ISA | 0.0619 | 0.0602 | **0.6043** | **0.6051** |
| VGG16 | IG | 0.0249 | 0.0232 | 0.0973 | 0.0959 |
| | FIG | 0.0288 | 0.0270 | 0.0809 | 0.0793 |
| | GIG | 0.0191 | 0.0173 | 0.1040 | 0.1025 |
| | BIG | 0.0390 | 0.0372 | 0.2274 | 0.2266 |
| | SM | 0.0434 | 0.0417 | 0.2710 | 0.2703 |
| | AGI | 0.0459 | 0.0442 | 0.4303 | 0.4304 |
| | ISA | 0.0504 | 0.0488 | **0.5111** | **0.5114** |

As shown in Table 1, the ISA method achieves the best attribution results with the highest Insertion score, which indicates that the ISA method outperforms other attribution methods for the attribution task. Specifically, the increase in Insertion score of the ISA method compared to other attribution methods is relatively large, with average increases of 0.3347 and 0.3544, and 0.3051 on Inception_v3, ResNet_50, and VGG16, respectively, indicating that the method has significantly improved the attribution performance.

## 5.5 ATTRIBUTION COMPLEXITY ANALYSIS

Regarding the attribution efficiency (time cost), it is typical to evaluate via the number of forward and back propagation times, such as in AGI(Pan et al., 2021). Following this way, we firstly consider the computational complexity of ISA is:

$$\left\lceil \frac{n}{k} \right\rceil \cdot (T_1 + T_2) \tag{11}$$

where $\lceil \cdot \rceil$ denotes the rounding up function. In our experiments, $T_1 = 8$ and $T_2 = 8$ represent the steps for gradient ascent and gradient descent. For the total feature number $n$ ($224 \times 224 \times 3$), we will attribute as large as possible $k$ features at a time (in our submission, $k = 5000$). Eventually, $\left\lceil \frac{n}{k} \right\rceil$ is about 30. For the AGI method, the computational complexity becomes $k \cdot m$, where $k$ is the number of false classes sampled, and $m$ is the maximum number of iterations. In (Pan et al., 2021), AGI takes $k = 20, m = 20$ on InceptionV3 respectively. Thus, our method demonstrates a huge performance improvement whilst attributing $k = 5000$ per attribution, although the time takes slightly longer than AGI (AGI is propagated 400 times and ISA is propagated 480 times). Overall,

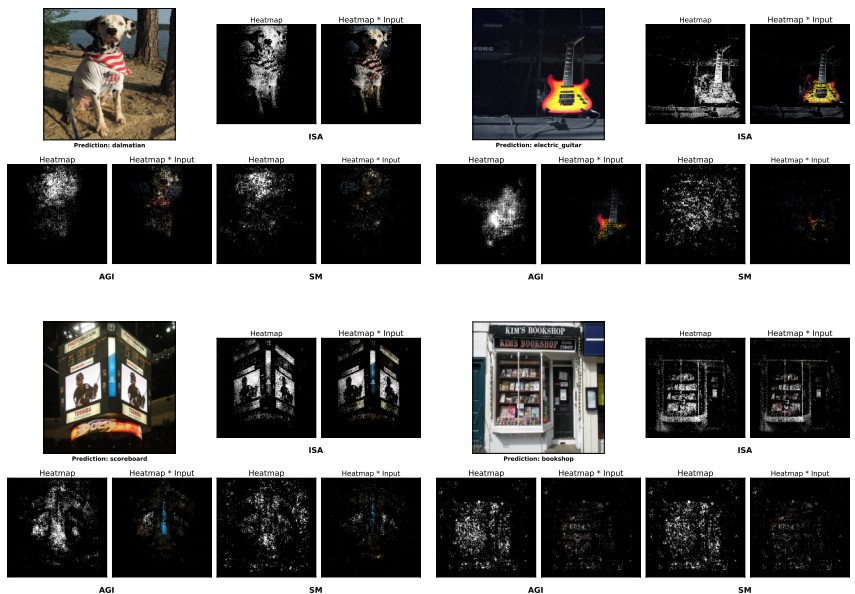

Figure 2: Attribution Results for *Scoreboard* Image using ISA, AGI, and SM

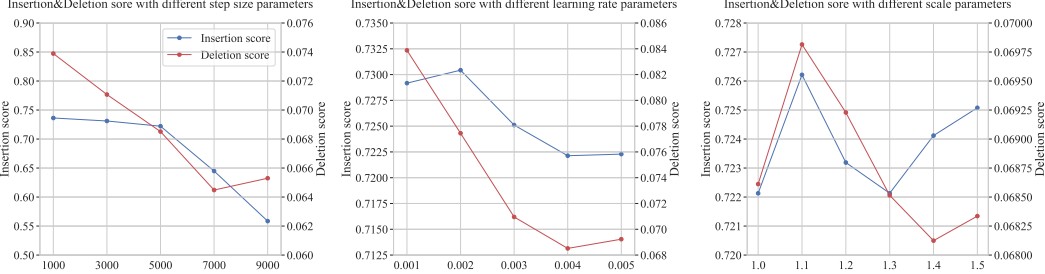

Figure 3: Insertion and Deletion score comparison of ISA

we consider the computational complexity of our algorithm to be reasonably comparable despite the iterative attribution nature, which could be very close to the AGI method.

## 5.6 ABLATION STUDY

In order to validate the efficacy of the ISA method, a series of ablation study is conducted using the inception_v3 model. These experiments aim to investigate the impact of various parameters on the model's performance. Specifically, we explore the effects of combining parameters of ascent step $T_1$ and descent step $T_2$, the effects of parameter step size, the effects of the parameter learning rate, and the effects of the parameter $S$. In the **appendix**, we additionally provide the ablation performance of ISA on VGG_16 and ResNet_50 models.

### 5.6.1 THE EFFECTS OF ASCENT STEP $T_1$ AND DESCENT STEP $T_2$

In this section, we compare the effects of two different approaches, gradient ascent and gradient descent, on the method's attribution performance. To acieve this, we set the following parameters: step size at 5000, ascent and descent steps at 8, learning rate at 0.004, and $S$ at 1.3. Three sets of experiments are conducted: gradient descent only, gradient ascent only, and simultaneous gradient descent and ascent. The combinations of these parameters are summarized in Table 2.

In the gradient descent-only and gradient ascent-only experiments, we observe similar values for the Insertion score and Deletion score. This similarity indicates that the attribution effects of these two

Table 2: Insertion sore and deletion score with different gradient parameters

|  | $T_1$ | $T_2$ | Insertion score | Deletion score |
|---|---|---|---|---|
| Gradient descent only | 0 | 8 | 0.7042 | 0.0715 |
| Gradient ascent only | 8 | 0 | 0.7055 | 0.0739 |
| Gradient descent and ascent | 8 | 8 | **0.7221** | **0.0685** |

methods are comparable. However, when gradient descent and ascent are performed simultaneously, the experimental results exhibit a higher Insertion score and a lower Deletion score. This suggests that the parameter combination used in the simultaneous approach outperforms the comparison experiments in terms of attribution effect. Specifically, the higher Insertion score combined with the lower Deletion score indicates the superiority of this particular parameter combination.

### 5.6.2 THE EFFECTS OF PARAMETER STEP SIZE

In this experiment, we conduct a comparison of the effect of different step sizes on the performance of ISA. Initially, we set ascent step $T_1$ and descent step $T_2$ to 8, learning rate to 0.004, and $S$ to 1.3. We tested step sizes of 1000, 3000, 5000, 7000, and 9000, respectively. The results are presented in Figure 3. From the figure, we can observe a decreasing trend in both the Insertion score and Deletion score as the step size increases. Specifically, when the step size is set to 1000, the Insertion score reaches its maximum value across all experiment results. Conversely, as the step size is increased to 9000, the Insertion score will be the lowest, accompanied by an increase in the Deletion score. Notably, when the step size was set to 5000, the Insertion score only decrease slightly compared to the maximum, while the Deletion score exhibits a significant decrease. Therefore, in our experiment, we set the step size to be 5000.

### 5.6.3 THE EFFECTS OF THE PARAMETER LEARNING RATE

In this experiment, we conduct a comparison of the attribution effects of ISA using different learning rates. The parameters are set as follows: ascent step $T_1$ and descent step $T_2$ at 8, step size at 5000, and $S$ at 1.3. Subsequently, we evaluate the performance of ISA at learning rates of 0.001, 0.002, 0.003, 0.004, and 0.005, respectively. The results are presented in Figure 3. As depicted in the figure, we can see that ISA achieves the highest Insertion score and competitively low Deletion score when the learning rate is 0.002. Both the Insertion score and Deletion score decline dramatically while the learning rates is increased. In our experiments, the learning rate is set to 0.002.

### 5.6.4 THE EFFECTS OF THE PARAMETER $S$

In this section, we conduct a performance comparison of ISA using different scales. The experimental setup involves setting ascent step $T_1$ and descent step $T_2$ to 8, step size to 5000, and learning rate to 0.04. We then test six different $S$: 1.0, 1.1, 1.2, 1.3, 1.4, and 1.5. The results are depicted in Figure 3. From the figure, we can see that both the Insertion score and Deletion score exhibit similar trends as $S$ increases. Therefore, we select the parameter with the highest Insertion score as the optimal choice. In this case, ISA achieve the best performance with a scale of 1.3.

## 6 CONCLUSION

In this paper, we propose a novel attribution method, called Iterative Search Attribution (ISA) to better interpret deep neural networks. Specifically, we consider that both gradient ascent and gradient descent are important for the exploration of sample importance, and we clip relatively unimportant features for the model to achieve more accurate attribution results. Comprehensive experimental results show that our method has superior performance for image recognition interpretability tasks compared with other state-of-the-art baselines. Given the limitation that we only explore the iterative attribution value by removing features in equal amounts, we will investigate the performance of our algorithm by removing features in unequal amounts, as the future work.

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
