# OpenReview forum: "Iterative Search Attribution for Deep Neural Networks"
_ICLR.cc/2024/Conference — ICLR 2024 Conference Withdrawn Submission_

### Official Review · Reviewer_VN3Q · 2023-10-29

**Soundness:** 3 good
**Presentation:** 4 excellent
**Contribution:** 3 good
**Rating:** 6
**Confidence:** 4

**Summary:**

To ensure the reliability of a DNN model and achieve the trustworthiness of deep learning, it is critical to enhance the interpretability and explainability of deep neural networks. Attribution methods are effective means of Explainable Artificial Intelligence (XAI) research. This paper, inspired by the iterative generation of Diffusion models, proposes an iterative search attribution (ISA) method by capturing the importance of samples during gradient ascent and descent and clipping the unimportant features in the model.  The method achieves SOTA performance.

**Strengths:**

1. The paper is well-written, with good organization and expressions.
2. The paper has the corresponding code released.
3. The achieved experimental results are State-of-the-art.
4. The justification of the proposed method, i.e., ISA, with both gradient descent and ascent considered is reasonable.
5. The authors perform detailed ablation studies to validate the proposed methodology.

**Weaknesses:**

1. How the paper is related to the diffusion models is unclear. Diffusion models apply iterative sampling whilst this paper is more on searching.
2. Though the method achieved good experimental results regarding the performance, and the qualitative evaluation, its efficiency is not largely improved, due to the iterative nature of the algorithm.
3. From Figure 3, it seems that the step size, learning rate, and scale parameters are quite sensible regarding the performance, any explanation for this?

**Questions:**

1.  Could discuss more on why iterative search is good.
2. Figure 3 shows several parameters are sensitive, please provide explanations.
3. The efficiency of the method is not improved a lot, which should be discussed, especially regarding the iterative method.

---

> ### Author Response · Authors · 2023-11-20
> **Response to Weaknesses**
>
> **Weaknesses:**
>
> **1.** Our method satisfies the property of autoregression, and diffusion models also satisfy it. Both our method and diffusion models iterate from their own state to obtain the best performance, because one-time generation often has poor quality, as evidenced by the relationship between VAE and Diffusion models.
>
> **2.** We fully acknowledge the viewpoint that the use of iterative methods can reduce algorithm efficiency.
>
> * In Section 5.5, we conducted a detailed analysis of the operational efficiency between our method and the state-of-the-art AGI. It is noteworthy that our algorithm achieves significant performance gains even when operating at a comparable efficiency to AGI (with 480 iterations of gradient propagation compared to AGI's 400 iterations).
>
> * We posit that, despite the efficiency gap arising from a slightly higher number of gradient propagations, the performance breakthrough achieved by our algorithm is a noteworthy outcome. This is why we prefer to use diffusion models to generate high-quality samples. Although we know that diffusion models usually take longer than non-diffusion models, we are actually willing to bear these efficiency reductions to obtain higher performance.
>
> **3.** Please see the following discussion for the clarification. We will update the explanations and motivations for introducing each parameter in the ablation experiments.
>
> * In Section 5.6.1 we conducted ablation experiments on the steps $T_{1}$ and $T_{2}$ of gradient ascent and gradient descent. These two parameters affect how well gradient ascent and gradient descent explore the input space as we described in **Weekness 1**. The higher the $T_{1}$ and $T_{2}$ values represent the deeper the exploration of the input space. In this article, we do not use different $T_{1}$ and $T_{2}$ for additional ablation, because a higher degree of exploration often means higher attribution accuracy, but it will increase the algorithm running time. , which is consistent with the intuition. We conducted ablation experiments on gradient ascent alone, gradient descent alone, and a combination of gradient ascent and gradient descent. The experimental results show that using both gradient ascent and gradient descent for input space exploration can achieve the highest performance.
>
> * In Section 5.6.2, the parameter ‘STEP SIZE’ means the number of unimportant attribution values to be removed in each iteration. A larger ‘STEP SIZE’ means more attribution values are removed in each iteration. We found that when ‘STEP SIZE’ is 5000, the algorithm achieves the best results. We believe that when 'STEP SIZE' is too low, the model may not be able to fully capture the contribution of different attribution values to the model's decision-making behavior. When 'STEP SIZE' is too high, noise may be introduced, some of which may be information irrelevant to the model, leading to inaccuracy in model interpretation.
>
> * The ‘learning rate’ in Section 5.6.3 is the learning rate corresponding to gradient ascent and gradient descent. Similar to the parameters $T_{1}$ and $T_{2}$ in Section 5.6.1, the learning rate affects the exploration process of the input space by gradient ascent and gradient descent. For gradient ascent, on the one hand, too high a learning rate may lead to over-exploration of the input space, making the interpretation too unstable. On the other hand, a learning rate that is too low may lead to an overly conservative exploration of the input space and miss potentially useful information. For gradient descent, on the one hand, a too high learning rate may cause the interpretation results to be too sensitive. On the other hand, a learning rate that is too low may increase the computational cost of interpreting results and limit the interpretation results to certain features in the input space.
>
> * The parameter S in Section 5.6.4 corresponds to our scaling factor in Section 4.4. Obviously, this parameter will affect the accuracy of our attribution. If the value of S is too large (close to 2), the corresponding importance of the attribution values removed in adjacent iterations will be closer, that is, it will be over-interming. Similarly, if the value of S is too small, it means under-interming, the estimation needs to be more accurate, and the attribution value removed in each iteration is not as good as the attribution value removed in the next iteration.

---

> ### Author Response · Authors · 2023-11-20
> **Response to Questions**
>
> **Questions:**
>
> **1.** We appreciate the constructive suggestion provided by the reviewer. The forthcoming response aims to supplement our algorithm with a comprehensive theoretical analysis.
>
> * We can start with a discussion of "local" versus "global" interpretability approaches. The "local" and "global" properties of attribution methods are relative to the input space. A local attribution method interprets the neural network's decisions within a specific neighborhood of an input data point $x_{0}$. It focuses on understanding the model's behavior in the vicinity of a particular instance rather than considering the entire input space. In contrast, a global attribution method assesses the importance of features across the entire range of possible inputs, considering the full input space. It provides insights into how different features contribute to the model's decisions on a broader scale.
>
> * So for conventional gradient-based interpretability methods, such as saliency maps [1] or Integrated Gradients [2], are often considered local because they operate based on the gradients computed at a specific anchor point or input instance. AGI [3] integrates the gradients from adversarial examples to the target example along the curve of steepest ascent to calculate the resulting contributions from all input features. Therefore, AGI does not rely on the selection of specific anchor points or input instances. It can calculate attributions over the entire input space and synthesize these attributions to provide a more global explanation.
>
> * Compared with AGI, our algorithm does not only consider the situation when gradient rises. We believe that gradient descent is also an important approach to explore the input space of deep neural networks. Therefore, we use samples after both gradient ascent and gradient descent exploration as baseline points for attribution to provide a more global interpretability explanation. This is the main innovation of our algorithm in Section 4.1. In Section 4.2, we discuss in detail which attribution results are less important when exploring the input space. After considering gradient ascent and gradient descent, we believe that smaller attribution values account for lower importance (we will reply to this argument in detail in global comment). Therefore, we creatively use the iterative attribution method in Sections 4.3 and 4.4 to remove unimportant smaller attribution values in each iteration. We use $max(attr_\gamma) < min(attr_{\gamma+1 })$ to constrain this removal. Also, importantly, we apply a scaling factor to ensure that the best attribution values in the previous iteration will perform worse than the worst attribution values in the latter iteration. Since each time we remove $k$ attribution values, we will normalize the remaining attribution results to between 0 and 1, so after multiplying by the scaling factor, a portion of the attribution values of neighboring iterations can be intermingled.
>
> **2.** Please refer to **Weakness 3**.
>
> **3.** Please refer to **Weakness 2**.
>
> References:
>
> [1] Simonyan, K., Vedaldi, A., & Zisserman, A. (2014, April). Deep inside convolutional networks: visualising image classification models and saliency maps. In Proceedings of the International Conference on Learning Representations (ICLR). ICLR.
>
> [2] Sundararajan, M., Taly, A., & Yan, Q. (2017, July). Axiomatic attribution for deep networks. In International conference on machine learning (pp. 3319-3328). PMLR.
>
> [3] Pan, D., Li, X., & Zhu, D. (2021, August). Explaining deep neural network models with adversarial gradient integration. In Thirtieth International Joint Conference on Artificial Intelligence (IJCAI).

---

### Official Review · Reviewer_8Joh · 2023-10-31

**Soundness:** 3 good
**Presentation:** 2 fair
**Contribution:** 2 fair
**Rating:** 6
**Confidence:** 3

**Summary:**

This paper proposes an Iterative Search Attribution (ISA) method to enhance the interpretability of deep neural networks (DNNs) by improving the accuracy of attribution. The authors introduce a scale parameter during the iterative process to ensure that the parameters in the next iteration are always more significant than the parameters in the current iteration. Experimental results demonstrate that the ISA method outperforms other state-of-the-art baselines in image recognition interpretability tasks.

**Strengths:**

The key contributions are laid out lucidly, helping readers discern the paper's main takeaways.
The method clips relatively unimportant features, leading to more accurate attribution results.

**Weaknesses:**

1. The comparison methods are not novel enough. There are some newer methodscan be compared(e.g. Explain Any Concept: Segment Anything Meets Concept-Based Explanation).
2. The experiments are relatively simple and do not adequately demonstrate the importance of the method in enhancing model interpretability.
3. The absence of corresponding theoretical analysis makes the method less convincing.

**Questions:**

See above.

---

> ### Author Response · Authors · 2023-11-20
> **Response to Weaknesses**
>
> **Weaknesses:**
>
> **1.** We appreciate the valuable suggestion from the reviewer. Our comparative experiments include several state-of-the-art attribution-based interpretability methods widely recognized in current computer vision tasks, such as IG [1], FIG [2], AGI [3], BIG [4], GIG [5]. We will incorporate an additional discussion on 'Segment Anything Meets Concept-Based Explanation' to enrich the argumentation of our experiments.
>
> **2.** We thank the reviewer for the suggestion. Our experiments meticulously adhere to the experimental setups of the current **SOTA methods AGI and BIG**, both of which employ Insertion score and Deletion score as quantitative metrics for evaluating interpretability. Our method has demonstrated performance far surpassing the current SOTA baselines. Moreover, as evidenced in Section 5.5, despite the utilization of iterative attribution, the efficiency of our algorithm is only slightly lower than that of AGI. Similar to diffusion models, our approach leverages the autoregressive nature to achieve outstanding performance while maintaining favorable efficiency.
>
> **3.** We appreciate the constructive suggestion provided by the reviewer. The forthcoming response aims to supplement our algorithm with a comprehensive theoretical analysis.
>
> * We can start with a discussion of "local" versus "global" interpretability approaches. The "local" and "global" properties of attribution methods are relative to the input space. A local attribution method interprets the neural network's decisions within a specific neighborhood of an input data point $x_{0}$. It focuses on understanding the model's behavior in the vicinity of a particular instance rather than considering the entire input space. In contrast, a global attribution method assesses the importance of features across the entire range of possible inputs, considering the full input space. It provides insights into how different features contribute to the model's decisions on a broader scale.
>
> * So for conventional gradient-based interpretability methods, such as saliency maps [6] or Integrated Gradients [1], are often considered local because they operate based on the gradients computed at a specific anchor point or input instance. AGI [3] integrates the gradients from adversarial examples to the target example along the curve of steepest ascent to calculate the resulting contributions from all input features. Therefore, AGI does not rely on the selection of specific anchor points or input instances. It can calculate attributions over the entire input space and synthesize these attributions to provide a more global explanation.
>
> * For our algorithm, it does not only consider the situation when gradient rises. We believe that gradient descent is also an important approach to explore the input space of deep neural networks. Therefore, we use samples after both gradient ascent and gradient descent exploration as baseline points for attribution to provide a more global interpretability explanation. This is the main innovation of our algorithm in Section 4.1. In Section 4.2, we discuss in detail which attribution results are less important when exploring the input space. After considering gradient ascent and gradient descent, we believe that smaller attribution values account for lower importance (we will reply to this argument in detail in global comment). Therefore, we creatively use the iterative attribution method in Sections 4.3 and 4.4 to remove unimportant smaller attribution values in each iteration. We use $max(attr_\gamma) < min(attr_{\gamma+1 })$ to constrain this removal. Also, importantly, we apply a scaling factor to ensure that the best attribution values in the previous iteration will perform worse than the worst attribution values in the latter iteration. Since each time we remove $k$ attribution values, we will normalize the remaining attribution results to between 0 and 1, so after multiplying by the scaling factor, a portion of the attribution values of neighboring iterations can be intermingled.

---

> ### Author Response · Authors · 2023-11-20
> **(Continued) Response to Weakness**
>
> References:
>
> [1] Sundararajan, M., Taly, A., & Yan, Q. (2017, July). Axiomatic attribution for deep networks. In International conference on machine learning (pp. 3319-3328). PMLR.
>
> [2] Hesse, R., Schaub-Meyer, S., & Roth, S. (2021). Fast axiomatic attribution for neural networks. Advances in Neural Information Processing Systems, 34, 19513-19524.
>
> [3] Pan, D., Li, X., & Zhu, D. (2021, August). Explaining deep neural network models with adversarial gradient integration. In Thirtieth International Joint Conference on Artificial Intelligence (IJCAI).
>
> [4] Wang, Z., Fredrikson, M., & Datta, A. (2022, June). Robust Models Are More Interpretable Because Attributions Look Normal. In International Conference on Machine Learning (pp. 22625-22651). PMLR.
>
> [5] Kapishnikov, A., Venugopalan, S., Avci, B., Wedin, B., Terry, M., & Bolukbasi, T. (2021). Guided integrated gradients: An adaptive path method for removing noise. In Proceedings of the IEEE/CVF conference on computer vision and pattern recognition (pp. 5050-5058).
>
> [6] Simonyan, K., Vedaldi, A., & Zisserman, A. (2014, April). Deep inside convolutional networks: visualising image classification models and saliency maps. In Proceedings of the International Conference on Learning Representations (ICLR). ICLR.

---

> > ### Comment · Reviewer_8Joh · 2023-11-21
> >
> > I have carefully reviewed all the comments from the reviewers and the authors' responses. I appreciate this work. However, I will not adjust the score until the authors have genuinely implemented 'We will' and updated the revision.

---

> > > ### Author Response · Authors · 2023-11-21
> > > **Response to EAC paper**
> > >
> > > Thanks for the follow-up.
> > >
> > > We appreciate the note of discussion of the paper 'Explain Any Concept: Segment Anything Meets Concept-Based Explanation'. To action on this implementation, we have updated the section for an enriched discussion as following (it has been included in our new version paper):
> > >
> > > =========================
> > >
> > > **2 RELATED WORK**
> > >
> > > **2.1 LOCAL APPROXIMATION MEHTODS TO INTERPRET DNNS**
> > >
> > > ....
> > >
> > > As one latest work on local approximation, `Explain Any Concept' (EAC) [1] uses Segment Anything Model (SAM) to perform accurate instance segmentation of images, and creatively proposes a lightweight per-input equivalent (PIE) scheme to reduce the running cost of the model. It is worth noting that in [1], EAC employs SAM and Shapley value for XAI. In our work, we primarily focus on the gradient-based attribution methods for interpretation. We will detail it in the following.
> > >
> > > ....
> > >
> > > =========================
> > >
> > > Ref:
> > >
> > > [1] Ao Sun, Pingchuan Ma, Yuanyuan Yuan, and Shuai Wang. Explain any concept: Segment anything meets concept-based explanation. arXiv preprint arXiv:2305.10289, 2023.

---

> > > ### Author Response · Authors · 2023-11-22
> > > **Response to theoretical analysis**
> > >
> > > For the theoretical analysis, we have updated the detailed proof as following (it has been included in our new version paper):
> > >
> > > =========================
> > >
> > > **Detailed proofs of the axiom of Sensitivity**
> > >
> > > Firstly, during the iterative process, the changes in the gradient along the integration path are captured by the original input information. Furthermore, it is not retroactive since feature values in previous iterations are unchanged in subsequent iterations. Therefore, the attribution result must be non-zero, which meets the definition of sensitivity. Here is the mathematical proof.
> > >
> > > We first use the first-order Taylor approximation to expand the loss function and combine the information for the path from $x_{0}$ to $x_{T}$.
> > >
> > > **Eq. 1:**
> > > $L\\left(x_{t}\\right) =L\\left(x_{t-1}\\right)\\pm\\frac{\\partial L\\left(x_{t-1}\\right)}{\\partial x_{t-1}}\\left(x_t-x_{t-1}\\right)+\\varepsilon$
> > >
> > > $\\sum_{t=1}^T L\\left(x_t\\right)  =\\sum_{t=0}^{T-1} L\\left(x_t\\right)\\pm\\sum_{t=0}^{T-1} \\frac{\\partial L\\left(x_t\\right)}{\\partial x_t}\\left(x_{t+1}-x_t\\right)$
> > >
> > > $A=L\\left(x_T\\right)- L\\left(x_0\\right)=\\pm\\sum_{t=0}^{T-1} \\frac{\\partial L\\left(x_t\\right)}{\\partial x_t}\\left(x_{t+1}-x_t\\right) = \\pm\\sum_{t=0}^{T-1} \\frac{\\partial L\\left(x_t\\right)}{\\partial x_t}\\cdot \\triangle x_{t}=\\pm\\int_{T} \\triangle x_{t}\\cdot \\frac{\\partial L\\left(x_t\\right)}{\\partial x_t}\\mathrm{d}t$
> > >
> > > Here $\epsilon$ is omitted due to the principle of higher-order Taylor expansions. $L$ represents the loss function. $x_{t}$ represents the input of the $t$-th iteration.
> > >
> > > We can know that as long as the loss function of the model changes, the attribution result will definitely be non-zero.
> > >
> > > Since our ISA algorithm combines gradient ascent and gradient descent to explore the input space, the ISA attribution path can be expanded as follows:
> > >
> > >    * For the input space exploration of gradient ascent, the attribution path of ISA is $x_{0}$, $x_{1}$,..., $x_{T_{1}}$.
> > >
> > >    * For the input space exploration of gradient descent, the attribution path of ISA is $x_{0}$, $x_{1}$,..., $x_{T_{2}}$.
> > >
> > > For the gradient ascent process, we can get the following formula:
> > >
> > > **Eq. 2:** $A_{a}=L\\left(x_{T_1}\\right)- L\\left(x_0\\right)=\\sum_{t=0}^{T_{1}-1} \\frac{\\partial L\\left(x_t\\right)}{\\partial x_t}\\left(x_{t+1}-x_t\\right) =\\sum_{t=0}^{T_{1}-1} \\frac{\\partial L\\left(x_t\\right)}{\\partial x_t}\\cdot \\triangle x_{t}=\\int_{T_{1}} \\triangle x_{t}\\cdot \\frac{\\partial L\\left(x_t\\right)}{\\partial x_t}\\mathrm{d}t $
> > >
> > > We define $\bigtriangleup L_{a}=L(x_{T_{1}})-L(x_{0})=c_{1}$. Here $c_{1}$ is a constant with a positive sign. Thus, we get $A_{ascent}=\frac{A_{a}}{\bigtriangleup L_{a}}$. According to Eq. 2, $A_{ascent}$ can be expressed as:
> > >
> > > **Eq. 3:**
> > > $A_{ascent}=\\frac{A_{a}}{\bigtriangleup L_{a}}=\frac{\sum_{t=0}^{T_{1}-1} \frac{\partial L\left(x_t\right)}{\partial x_t}\left(x_{t+1}-x_t\right)}{L(x_{T_{1}})-L(x_{0})}\\
> > >  =\frac{\sum_{t=0}^{T_{1}-1} \frac{\partial L\left(x_t\right)}{\partial x_t}\cdot \triangle x_{t}}{L(x_{T_{1}})-L(x_{0})}=\frac{1}{c_{1}}\int_{T_{1}} \triangle x_{t}\cdot \frac{\partial L\left(x_t\right)}{\partial x_t}\mathrm{d}t=1 $
> > >
> > > The attribution result is normalized to 1, which satisfies sensitivity. It is worth noting that since $c_1$ is a constant with a positive sign, during gradient ascent, we can use $L'=\frac{L}{c_{1}}$ to replace the loss function $L$ in Eq. 3, so sensitivity is also satisfied at this time.
> > >
> > > For the gradient descent process, similarly, we can get the following formula:
> > >
> > > **Eq. 4:**
> > > $A_{d}=L\left(x_{T_{2}}\right)-  L\left(x_{0}\right)=-\sum_{t=0}^{T_{2}-1} \frac{\partial L\left(x_t\right)}{\partial x_t}\left(x_{t+1}-x_t\right) =-\sum_{t=0}^{T_{2}-1} \frac{\partial L\left(x_t\right)}{\partial x_t}\cdot \triangle x_{t}=-\int_{T_{2}} \triangle x_{t}\cdot \frac{\partial L\left(x_t\right)}{\partial x_t}\mathrm{d}t $
> > >
> > > We define $\bigtriangleup L_{d}=L(x_{T_{2}})-L(x_{0})=c_{2}$. Here $c_{2}$ is a constant with a negative sign. Thus, we get $A_{descent}=\frac{A_{d}}{\bigtriangleup L_{d}}$. According to Eq. 4, $A_{descent}$ can be expressed as:
> > >
> > > **Eq. 5:** $A_{descent}=\frac{A_{d}}{\bigtriangleup L_{d}}=\frac{-\sum_{t=0}^{T_{2}-1} \frac{\partial L\left(x_t\right)}{\partial x_t}\left(x_{t+1}-x_t\right)}{L(x_{T_{2}})-L(x_{0})} =\frac{\sum_{t=0}^{T_{2}-1} \frac{\partial L\left(x_t\right)}{\partial x_t}\cdot \triangle x_{t}}{L(x_{0})-L(x_{T_{2}})}=\frac{1}{-c_{2}}\int_{T_{2}} \triangle x_{t}\cdot \frac{\partial L\left(x_t\right)}{\partial x_t}\mathrm{d}t=1$
> > >
> > > We can see that the attribution result is normalized to 1, which satisfies sensitivity. It is worth noting that since $c_2$ is a constant with a negatove sign, during gradient descent, we can use $L''=\frac{L}{-c_{2}}$ to replace the loss function $L$ in Eq. 5, so sensitivity is also satisfied.
> > >
> > > =========================

---

> > > ### Author Response · Authors · 2023-11-22
> > > **Response to theoretical analysis (Continued)**
> > >
> > > Finally, we make a balance between gradient ascent and descent by combining them in the following formula:
> > >
> > > $Attr=A_{ascent}+A_{descent}=\int_{T_{1}} \triangle x_{t}\cdot \frac{\partial L'\left(x_t\right)}{\partial x_t}\mathrm{d}t+\int_{T_{2}} \triangle x_{t}\cdot \frac{\partial L''\left(x_t\right)}{\partial x_t}\mathrm{d}t=2$
> > >
> > > We get $\frac{Attr}{2}=\frac{\int_{T_{1}} \triangle x_{t}\cdot \frac{\partial L'\left(x_t\right)}{\partial x_t}\mathrm{d}t+\int_{T_{2}} \triangle x_{t}\cdot \frac{\partial L''\left(x_t\right)}{\partial x_t}\mathrm{d}t}{2}=1$, which also satisfies sensitivity.
> > >
> > > ++++++++++++++++++++++++++++++++
> > >
> > > We have included the above section in our new version of paper.

---

### Official Review · Reviewer_df7s · 2023-11-01

**Soundness:** 2 fair
**Presentation:** 2 fair
**Contribution:** 2 fair
**Rating:** 3
**Confidence:** 4

**Summary:**

This paper proposes a new method for attributing a neural network prediction to input features. The proposed method integrates the gradient with respect to input features over both the gradient ascent and descent paths. The paper provides experiments on ImageNet where it shows better performance compared to prior methods under Insertion score.

**Strengths:**

The paper tackles an interesting problem, namely providing better saliency maps. It proposes interesting and novel modifications to existing gradient-based attribution methods, and seems to make some improvement upon existing methods.

**Weaknesses:**

1- The paper makes many ad-hoc design choices which are not well justified. The provided explanations seem unclear to me (Section 4.2). It is helpful to provide experiments that show how each design choice affects the attribution, as well as more formal explanations (using clear mathematical notation).

2- While the ablation studies show which hyper-parameters were most useful, they lack explanation of why and clear connection to the motivation for introducing the respective parameters.

3- The metrics, Insertion and Deletion, are not formally defined. Since these two metrics are the basis of the main results in Table 1, it is important to clarify their definition and justify their meaningfulness. It is particularly useful to show what is the real-world consequences of lower Insertion or Deletion.

4- The paper mentions that Grad-CAM and Score-CAM perform poorly in non-CNN models, however, all the results in the paper are reported with CNN based models, so a comparison to these methods are required.

5- The paper can improve in writing (also Figure 1 is not clear to me). Please make sure claims are either backed by specific citations or experiments, for example: “Unfortunately, these two methods are more suitable for CNNs and perform poorly in non-CNN cases”.

6- minor typo: Section 4.2 must be \Delta x_k not x_t

**Questions:**

My suggestions are included in the issues I raised in the weaknesses section.

---

> ### Author Response · Authors · 2023-11-20
> **Response to Weaknesses**
>
> **Weaknesses:**
>
> **1.** To address the concern, we herein provide a comprehensive clarification regarding the design of our algorithm.
>
> * We start with a discussion of "local" versus "global" interpretability approaches. The "local" and "global" properties of attribution methods are relative to the input space. A local attribution method interprets the neural network's decisions within a specific neighborhood of an input data point $x_{0}$. It focuses on understanding the model's behavior in the vicinity of a particular instance rather than considering the entire input space. In contrast, a global attribution method assesses the importance of features across the entire range of possible inputs, considering the full input space. It provides insights into how different features contribute to the model's decisions on a broader scale.
>
> * For conventional gradient-based interpretability methods, such as saliency maps [1] or Integrated Gradients [2], are often considered local because they operate based on the gradients computed at a specific anchor point or input instance. AGI [3] integrates the gradients from adversarial examples to the target example along the curve of steepest ascent to calculate the resulting contributions from all input features. Therefore, AGI does not rely on the selection of specific anchor points or input instances. It can calculate attributions over the entire input space and synthesize these attributions to provide a more global explanation.
> * For our algorithm, it does not only consider the situation when gradient rises. We believe that gradient descent is also an important approach to explore the input space of deep neural networks. Therefore, we use samples after both gradient ascent and gradient descent exploration as baseline points for attribution to provide a more global interpretability explanation.
>
> * This is the main innovation of our algorithm in Section 4.1. In Section 4.2, we discuss in detail which attribution results are less important when exploring the input space. After considering gradient ascent and gradient descent, we believe that smaller attribution values account for lower importance (we will reply to this argument in detail in global comment). Therefore, we creatively use the iterative attribution method in Sections 4.3 and 4.4 to remove unimportant smaller attribution values in each iteration. We use $max(attr_\gamma) < min(attr_{\gamma+1 })$ to constrain this removal. Also, importantly, we apply a scaling factor to ensure that the best attribution values in the previous iteration will perform worse than the worst attribution values in the latter iteration. Since each time we remove $k$ attribution values, we will normalize the remaining attribution results to between 0 and 1, so after multiplying by the scaling factor, a portion of the attribution values of neighboring iterations can be intermingled.

---

> ### Author Response · Authors · 2023-11-20
> **(Continued) Response to Weakness**
>
> **2.** Thank you for the constructive suggestions! We will add relevant explanations and motivations for introducing each parameter in the ablation experiments. Please first allow me to provide some clarification here.
>
> * In Section 5.6.1 we conducted ablation experiments on the steps $T_{1}$ and $T_{2}$ of gradient ascent and gradient descent. These two parameters affect how well gradient ascent and gradient descent explore the input space as we described in **Weekness 1**. The higher the $T_{1}$ and $T_{2}$ values represent the deeper the exploration of the input space. In this article, we do not use different $T_{1}$ and $T_{2}$ for additional ablation, because a higher degree of exploration often means higher attribution accuracy, but it will increase the algorithm running time. , which is consistent with the intuition. We conducted ablation experiments on gradient ascent alone, gradient descent alone, and a combination of gradient ascent and gradient descent. The experimental results show that using both gradient ascent and gradient descent for input space exploration can achieve the highest performance.
>
> * In Section 5.6.2, the parameter ‘STEP SIZE’ means the number of unimportant attribution values to be removed in each iteration. A larger ‘STEP SIZE’ means more attribution values are removed in each iteration. We found that when ‘STEP SIZE’ is 5000, the algorithm achieves the best results. We believe that when 'STEP SIZE' is too low, the model may not be able to fully capture the contribution of different attribution values to the model's decision-making behavior. When 'STEP SIZE' is too high, noise may be introduced, some of which may be information irrelevant to the model, leading to inaccuracy in model interpretation.
> * The ‘learning rate’ in Section 5.6.3 is the learning rate corresponding to gradient ascent and gradient descent. Similar to the parameters $T_{1}$ and $T_{2}$ in Section 5.6.1, the learning rate affects the exploration process of the input space by gradient ascent and gradient descent. For gradient ascent, on the one hand, too high a learning rate may lead to over-exploration of the input space, making the interpretation too unstable. On the other hand, a learning rate that is too low may lead to an overly conservative exploration of the input space and miss potentially useful information. For gradient descent, on the one hand, a too high learning rate may cause the interpretation results to be too sensitive. On the other hand, a learning rate that is too low may increase the computational cost of interpreting results and limit the interpretation results to certain features in the input space.
>
> * The parameter $S$ in Section 5.6.4 corresponds to our scaling factor in Section 4.4. Obviously, this parameter will affect the accuracy of our attribution. If the value of S is too large (close to 2), the corresponding importance of the attribution values removed in adjacent iterations will be closer, that is, it will be over-interming. Similarly, if the value of S is too small, it means under-interming, the estimation needs to be more accurate, and the attribution value removed in each iteration is not as good as the attribution value removed in the next iteration.

---

> ### Author Response · Authors · 2023-11-20
> **(Continued) Response to Weakness**
>
> **3.** We are grateful to the reviewer's suggestion. Here we provide specific definitions of Insertion Score and Deletion Score. Starting from a blank image, the insertion game successively inserts pixels from highest to lowest attribution scores and makes predictions. If we draw a curve that represents the prediction values, the area under the curve (AUC) is then defined as the insertion score. higher the insertion score, the better the quality of interpretation. Similarly, starting from the original image, the deletion score is obtained by successively deleting the pixels from the highest to lowest attribution scores. The lower the deletion score, the better the quality of interpretation. More detailed references can be found in [3],[4].
>
> **4.** For the point of view "Unfortunately, these two methods are more suitable for CNNs and perform poorly in non-CNN cases", we would like to refer to the **AGI paper [3]**, in which the third paragraph of the Introduction Section: 'Within gradient models, although CAM based methods give promising results in various applications, a major limitation is that it applies only to Convolutional Neural Network (CNN) architectures'.
>
>
> **5.** Thank you for the suggestion. We will update the writing of the paper and reconstruct Figure 1 to provide readers with a clearer understanding of our article. We will review the article and add appropriate reference citations. For the point of view "Unfortunately, these two methods are more suitable for CNNs and perform poorly in non-CNN cases", we would like to refer to the third paragraph of the Introduction Section in **AGI paper [3]**: 'Within gradient models, although CAM based methods give promising results in various applications, a major limitation is that it applies only to Convolutional Neural Network (CNN) architectures' .
> We express our gratitude for the time and efforts from the reviewers. We pledge to thoroughly review the article, rectify any identified issues, and take measures to prevent the recurrence of such errors.
>
> References:
>
> [1] Simonyan, K., Vedaldi, A., & Zisserman, A. (2014, April). Deep inside convolutional networks: visualising image classification models and saliency maps. In Proceedings of the International Conference on Learning Representations (ICLR). ICLR.
>
> [2] Sundararajan, M., Taly, A., & Yan, Q. (2017, July). Axiomatic attribution for deep networks. In International conference on machine learning (pp. 3319-3328). PMLR.
>
> [3] Pan, D., Li, X., & Zhu, D. (2021, August). Explaining deep neural network models with adversarial gradient integration. In Thirtieth International Joint Conference on Artificial Intelligence (IJCAI).
>
> [4] Petsiuk, V., Das, A., & Saenko, K. (2018). Rise: Randomized input sampling for explanation of black-box models. arXiv preprint arXiv:1806.07421.

---

> > ### Comment · Reviewer_df7s · 2023-11-22
> > **Remaining concerns**
> >
> > Thank you for your response. My 4th concern regarding comparing to Grad-CAM remains (since all the reported results are based on CNNs). I am also still not clear about the motivation, and Sections 4.2 remains hard to understand for me. Specially, I do not understand the logic in this argument in the global response: "For gradient descent, parameters with larger attribution values are unimportant. The reason is that, if we change these values, the gradient can still decrease and the model performance becomes better."

---

> > > ### Author Response · Authors · 2023-11-23
> > > **Clarification and additional experiments**
> > >
> > > Thank you for your reply.
> > >
> > > **Firstly**, to address your concern, we would like to clarify that, in the paper of AGI [1], the shortcomings of GradCAM in interpretability is discussed in the Introduction section:
> > >
> > > =============================================
> > >
> > > *Within gradient models, although CAM based methods give promising results in various applications, a major limitation is that it applies only to Convolutional Neural Network (CNN) architectures*
> > >
> > > =============================================
> > >
> > > This is also included in our rebuttal of **(Continued) Response to Weakness**.
> > >
> > > Nor to further empirically address the concern, we have included additional experiments comparing our method with GradCAM.
> > >
> > > In addition, we also conduct experiments on the Vision Transformer (ViT) model to verify the superiority of our method. The experimental results can be found in the global comment (Table 1 and Table 2).
> > >
> > > **Secondly**, for the logic of 'For gradient descent, parameters with larger attribution values are unimportant.' The reason is that, if we change these values, the gradient can still decrease and the model performance becomes better.', here is our explanation:
> > >
> > > For neural network input, we will use 0 as the missing feature, which is also the foundation of the attribution evaluation system. So if a feature changes to 0, which means a feature is removed, we will check the contribution of a feature from its original state to 0. If removing a feature can reduce the loss function, that is, improve the confidence, then this feature is not important and is irrelevant to obtaining this category. In the same way, in the gradient descent process of a feature, a larger attribute value means a greater contribution, and removing it also means it is unimportant.
> > >
> > > Ref:
> > >
> > > [1]. Pan, D., Li, X., & Zhu, D. (2021, August). Explaining deep neural network models with adversarial gradient integration. In Thirtieth International Joint Conference on Artificial Intelligence (IJCAI).

---

### Official Review · Reviewer_PAUU · 2023-11-04

**Soundness:** 2 fair
**Presentation:** 1 poor
**Contribution:** 2 fair
**Rating:** 3
**Confidence:** 3

**Summary:**

This paper proposed a gradient-based iterative attribution method, the Iterative Search Attribution (ISA), which combines gradient descent and gradient ascent to construct the integration path. Experiments demonstrate the effectiveness of the proposed ISA.

**Strengths:**

- the idea of combining both gradient descent (GD) and gradient ascent (GA) in constructing the integration path in the gradient-based attribution method is interesting (and, based on their results, useful).
- the qualitative results in Figure 2 look promising.

**Weaknesses:**

- the discussion about the local and global attribution methods is vague.
    - are "local" and "global" properties of the attribution method concerning the input space? namely, a local attribution method only interprets the NN within a neighborhood of an input $x_0$, while a global attribution method will assess the importance of features in $x_0$ with respect to the whole range of possible input (full input space), thus reflecting the property of the NN itself.
    - If so, one may argue that the gradient-based methods are also local as they depend on specific anchor points.
- the discussion about the relative importance of the features with "larger attribution" in GA and GD cases in Sec.4.2 is totally confusing and I fail to see connections to the algorithm
- the algorithm is not clearly described and has minor typos, e.g.,
    - $A_a$ is not introduced and initialized
    - lines 4 and 8: $x_t = x_t \dots$ should be $x_{t+1}=x_t \dots$
    - line 14: what does symbol $min_k$ mean? taking the lowest $k$ features from from $Attr_\gamma$ (then $attr_\gamma$ is a $k$-dim vector)?
- the constraints in Sec. 4.3 seem unnecessary and the theoretical reason for choosing them is not stated.  In addition, the introduced hyper-parameter $S$ could break the previous constraint, i.e., $max(attr_\gamma) < min(attr_{\gamma+1})$.
- the author should pay more effort to justify their claim that "the Insertion score serves as a more representative indicator of the performance of attribution algorithms." (at least should be more than just one paragraph!)
    - especially, it is interesting to note that the ISA always has the worst deletion score in Table 1

**Questions:**

- could you provide more concrete descriptions (and comparisons) about the local and global attribution methods?
- could rephrase your arguments in Sec. 4.2 to make it clear to understand?
- could you theoretically justify the proposed constraints stated in Sec. 4.3? Also, what if you do not add $\gamma$ to $attr_\gamma$?
- could you provide in-depth (better to be quantitative) arguments about the claim that "the Insertion score serves as a more representative indicator of the performance of attribution algorithms."?

---

> ### Author Response · Authors · 2023-11-20
> **Response to Weaknesses**
>
> **Weaknesses:**
>
> 1. For global attribution methods, it depends on the implementation details.
>
> * Local attribution method interprets the neural network's decisions within a specific neighborhood of an input data point $x_{0}$
>
> * Global attribution method assesses the importance of features across the entire range of possible inputs, considering the full input space. For conventional gradient-based interpretability methods, such as saliency maps [1] or Integrated Gradients [2], are often considered local because they operate based on the gradients computed at a specific anchor point or input instance.
>
> * However, AGI [3] integrates the gradients from adversarial examples to the target example along the curve of steepest ascent to calculate the resulting contributions from all input features. Therefore, AGI does not rely on the selection of specific anchor points or input instances. It can calculate attributions over the entire input space and synthesize these attributions to provide a more global explanation.
>
> * For our method, we further extend the capability of AGI as a global attribution method by considering gradient descent to the input space of deep neural networks. Therefore, we use samples after both gradient ascent and gradient descent exploration as baseline points for attribution to provide a more global interpretability explanation. This is the main innovation of our algorithm in Section 4.1. In Section 4.2, we discuss in detail which attribution results are less important when exploring the input space. After considering gradient ascent and gradient descent, we believe that smaller attribution values account for lower importance (we will reply to this argument in detail in **Global comment**). Therefore, we creatively use the iterative attribution method in Sections 4.3 and 4.4 to remove unimportant smaller attribution values in each iteration. We use $max(attr_\gamma) < min(attr_{\gamma+1 })$ to constrain this removal. Also, importantly, we apply a scaling factor to ensure that the best attribution values in the previous iteration will perform worse than the worst attribution values in the latter iteration. Since each time we remove $k$ attribution values, we will normalize the remaining attribution results to between 0 and 1, so after multiplying by the scaling factor, a portion of the attribution values of neighboring iterations can be intermingled.
>
> 2. Thanks for bringing this into our attention. We have provided the updated details in the **Global comment**. The new version will incorporate these details.
>
> 3. We promise to adjust the pseudocode to include initialization $A_{a}$ and correct errors on lines 4 and 8. Here, $min_{k}$ in line 14 means that the lowest $k$ features are taken.
>
> 4. That is exactly the design of our algorithm. Both constraints in Sections 4.3 and 4.4 are interrelated. The constraint in Section 4.3 is a strict constraint, while the constraint in Section 4.4 is an approximate one. The premise of establishing such a strict constraint in Section 4.3 is that the features that are removed first are less important features. But in fact, such a constraint may be too hard to the model, so we introduce the hyperparameter $S$ to break such strict constraints and ensure that the best attribution values in the previous iteration will perform worse than the worst attribution values in the latter iteration .
>
> 5. We agree with the reviewer on this point. However, it doesn’t affect the conclusion of this work. The reasons are from two folds:
>
>       * The difference between our method and other baselines on Inception-v3, ResNet50 and VGG16 are extremely small for the deletion score.
>
>       * We would like to highlight that the insertion score is a more important metric compared to the deletion score. Since the insertion score measures the degree of change in the output of the model when pixels are inserted into the input, it is the accumulation of accuracy from scratch. The deletion score represents the accumulation of accuracy of the remaining parts when features are deleted. Thus, the accuracy calculated by the deletion score is based on the feature deletion process. At this time, the undeleted features will interfere with the accuracy and affect the effect of the interpretability evaluation.
>       * To sum up, we believe that ‘the Insertion score serves as a more representative indicator of the performance of attribution algorithms’.
>
>  We will incorporate these discussions in Section 5.2.

---

> ### Author Response · Authors · 2023-11-20
> **Response to Questions**
>
> **Questions:**
>
> 1. Addressed in **Weakness 1**.
>
> 2. As discussed before in **Weakness** and **Global Comment**, we will add the updated discussion in Section 4.2 to provide readers with a clearer understanding of our main contributions.
>
> 3. As discussed in **Weakness 1**, we need to ensure that the $k$ attribution values removed in the previous iteration are always less important (i.e. smaller) than the attribution values removed in the later iteration. We use $max(attr_\gamma) < min(attr_{\gamma+1 })$ to limit this deletion. Since each time we remove $k$ attribution values, we will normalize the remaining attribution results to be between 0 and 1, so in order to mix a part of the attribution values from adjacent iterations together, ensuring the best attribution values in the previous iteration will perform worse than the worst attribution values in the latter iteration, we apply a scaling factor.
>
> Here, $\gamma$ refers to the iteration of $\gamma$. If $\gamma$ is not added to $attr_{\gamma}$, it will not be an iterative attribution.
>
> 4. Addressed in **Weakness 5**.
>
> References:
>
> [1] Simonyan, K., Vedaldi, A., & Zisserman, A. (2014, April). Deep inside convolutional networks: visualising image classification models and saliency maps. In Proceedings of the International Conference on Learning Representations (ICLR). ICLR.
>
> [2] Sundararajan, M., Taly, A., & Yan, Q. (2017, July). Axiomatic attribution for deep networks. In International conference on machine learning (pp. 3319-3328). PMLR.
>
> [3] Pan, D., Li, X., & Zhu, D. (2021, August). Explaining deep neural network models with adversarial gradient integration. In Thirtieth International Joint Conference on Artificial Intelligence (IJCAI).

---

### Author Response · Authors · 2023-11-20
**Further discussion of the Section 4.2**

At the beginning of Section 4.2, we introduced the process of gradient ascent and gradient descent to explore the input space through Equations 5 and 6. $A$ represents the sum of attributions, where the more accurate expression of '$\bigtriangleup x$' should be '$\bigtriangleup x_{t}$'. If we are conducting gradient ascent exploration, Equation 6 corresponds to $\bigtriangleup x_{t}=+ \eta \cdot sign(\frac{\partial L(x_{t};W)}{\partial x_{t }} )$. If we are conducting gradient descent exploration, Equation 6 corresponds to $\bigtriangleup x_{t}=- \eta \cdot sign(\frac{\partial L(x_{t};W)}{\partial x_{t }} )$.

Next, we conducted the following **Presentation** in Section 4.2:

Our goal is to confirm which parameters are important through attribution, because attribution can indicate the contribution of each parameter to the loss change. Therefore, we first let the loss change, and then use attribution to evaluate the contribution. Correspondingly, the methods of changing loss include gradient ascent and gradient descent, in which their direction function $sign$ are used in these two processes in order to change each parameter fairly and not be affected by the gradient value.

We believe that the importance of attribution values depends on whether their corresponding model parameters can be changed. Therefore, parameters with larger attribution values are important during gradient ascent. Since changing these parameters will cause the loss to increase greatly, it may impact the model performance. For gradient descent, parameters with larger attribution values are unimportant. The reason is that， if we change these values, the gradient can still decrease and the model performance becomes better.

With the above explanation, we can get the following two **conclusions**:

1. During gradient ascent, parameters with smaller attribution values are unimportant.

2. During gradient descent, parameters with larger attribution values are unimportant.

Since gradient ascent can only be obtained when destroying the model, and gradient ascent alone ignores the usefulness of gradient descent for exploring the input space, we combine the gradient ascent and gradient descent processes by using $A_{ascent}+A_{descent}$ (Eq. 7). Here $A_{ascent}=\frac{A}{\ bigtriangleup L_{a}}$ and $A_{descent}=\frac{A}{\bigtriangleup L_{d}}$.

Finally, since the value of $L_{a}$ is positive and the value of $L_{d}$ is negative, according to Equation 7, in *Discussion*, our **conclusion** changes to:

* **During gradient ascent, parameters with smaller attribution values are unimportant.**

* **During gradient descent, parameters with smaller attribution values are unimportant.**

It is obvious that when combining gradient ascent and gradient descent, parameters with smaller attribution values are unimportant. We used this perspective to conduct subsequent research on iterative attribution.

---

### Author Response · Authors · 2023-11-23
**Global comment for additional experiments**

Table1. Comparison of ISA and GradCAM
| 　 | Inception-v3 |  | ResNet-50 |  | VGG16 |  |
|:---:|:---:|:---:|:---:|:---:|:---:|:---:|
| Method | Insertion  | Deletion | Insertion  | Deletion | Insertion | Deletion |
| GradCAM | 0.5798 | 0.1594 | 0.3417 | 0.1231 | 0.4545 | 0.1092 |
| ISA | **0.7293** | **0.0745** | **0.6043** | **0.0619** | **0.5111** | **0.0504** |

Table 2. Experiments on ViT
| 　 | vit_b_16 |  |
|:---:|:---:|:---:|
| Method | Insertion  | Deletion |
| Fast-IG | 0.2682 | 0.0883 |
| Saliency Map | 0.3374 | 0.098 |
| IG | 0.3782 | 0.068 |
| GIG | 0.366 | 0.06306 |
| AGI | 0.5014 | 0.0849 |
| BIG | 0.4704 | 0.1132 |
| ISA  | **0.6715** | 0.1153 |